# The Efficacy of an N-Acetylcysteine–Antibiotic Combination Therapy on *Achromobacter xylosoxidans* in a Cystic Fibrosis Sputum/Lung Cell Model

**DOI:** 10.3390/biomedicines10112886

**Published:** 2022-11-10

**Authors:** Aditi Aiyer, Theerthankar Das, Gregory S. Whiteley, Trevor Glasbey, Frederik H. Kriel, Jessica Farrell, Jim Manos

**Affiliations:** 1Charles Perkins Centre, Infection, Immunity and Inflammation, Sydney Institute for Infectious Diseases, School of Medical Sciences, The University of Sydney, Sydney, NSW 2006, Australia; 2Whiteley Corporation, Level 5, 12 Mount Street North Sydney, Sydney, NSW 2060, Australia; 3School of Medicine, Western Sydney University, Sydney, NSW 2566, Australia; 4Whiteley Corporation, 19-23 Laverick Avenue, Tomago, NSW 2322, Australia

**Keywords:** cystic fibrosis, *Achromobacter xylosoxidans*, BEAS-2B, ASMDM-1, artificial sputum medium, N-acetylcysteine, cell/Sputum model

## Abstract

Cystic fibrosis (CF) is a disorder causing dysfunctional ion transport resulting in the accumulation of viscous mucus. This environment fosters a chronic bacterial biofilm-associated infection in the airways. *Achromobacter xylosoxidans*, a gram-negative aerobic bacillus, has been increasingly associated with antibiotic resistance and chronic colonisation in CF. In this study, we aimed to create a reproducible model of CF infection using an artificial sputum medium (ASMDM-1) with bronchial (BEAS-2B) and macrophage (THP-1) cells to test *A. xylosoxidans* infection and treatment toxicity. This study was conducted in three distinct stages. First, the tolerance of BEAS-2B cell lines and two *A. xylosoxidans* strains against ASMDM-1 was optimised. Secondly, the cytotoxicity of combined therapy (CT) comprising N-acetylcysteine (NAC) and the antibiotics colistin or ciprofloxacin was tested on cells alone in the sputum model in both BEAS-2B and THP-1 cells. Third, the efficacy of CT was assessed in the context of a bacterial infection within the live cell/sputum model. We found that a model using 20% ASMDM-1 in both cell populations tolerated a colistin–NAC-based CT and could significantly reduce bacterial loads in vitro (~2 log_10_ CFU/mL compared to untreated controls). This pilot study provides the foundation to study other bacterial opportunists that infect the CF lung to observe infection and CT kinetics. This model also acts as a springboard for more complex co-culture models.

## 1. Introduction

Cystic fibrosis (CF) is an autosomal recessive genetic disease that is caused by mutations in the CF transmembrane conductance regulator (CFTR) protein [1]. This results in dysfunctional transepithelial ion transport, which is critical to maintain appropriate ionic composition of the airway surface liquid (ASL). CF presents with dehydrated ASL and elevated bronchial mucus production [1]. This creates the ideal environment for opportunistic and progressively chronic bacterial colonisation. While the most common pathogen is *Pseudomonas aeruginosa* [2], in the past decade newer or emerging pathogens pose an increasing risk to patient treatment. One such pathogen is *Achromobacter xylosoxidans*.

*A. xylosoxidans* is a gram-negative aerobic bacillus that is ubiquitous in the environment. It has been associated with increased antibiotic resistance, lung inflammation in CF, and severe lung disease in adults [3,4]. The rate of colonisation of *A. xylosoxidans* varies between 2–17.9% worldwide; however, this figure could be underestimated as it is often confused with other pathogens such as *P. aeruginosa,* especially in laboratories that are unequipped to manage nuanced CF identification [5]. Our previous research has highlighted how the combination of the neutralised form of N-acetylcysteine (NAC) (neutralised form referred to herein as “NAC_neutral_”) enhances the effect of the antibiotics colistin and ciprofloxacin in certain clinical strains of *A. xylosoxidans* [6]. While it is understood through discriminative molecular tools that the presence of *A. xylosoxidans* is becoming more prevalent in recent years, current research lacks an understanding of chronic *Achromobacter* spp. colonisation and adaptation to the human lungs. Thus, further study is required to clarify its pathogenesis in CF lung disease [5].

Antibiotics are conventionally administered to manage CF bacterial infections. However, due to rising antibiotic resistance, different therapies or combined treatments need to be investigated. Recent studies have suggested that antibiotic activity may be improved by employing a combination therapy (CT) with antioxidants. NAC has been investigated against gram-negative opportunistic CF pathogens with promising results in vitro. Previous studies have investigated the related antioxidant, glutathione (GSH), on A549 cells following the addition of pyocyanin, which is a *P. aeruginosa*-released virulence factor [7]. NAC is recognised as a precursor to glutathione and so their activities are expected to be similar [8]. However, to our knowledge, NAC_neutral_ has not been investigated in a similar way to GSH in a reproducible model. 

While most cell lines have been derived from virally transformed tissue, recent studies indicate they are an ideal surrogate system to monitor drug metabolism, toxicity, and infectious pathology [9] and provide an ethical alternative to animal testing [10,11]. The pulmonary epithelia are defined according to their region of isolation; upper, lower, and distal. Primary and secondary manifestations of infection and inflammation in CF occur in the lower respiratory tract, the trachea, and primary bronchi. While the gold standard to conduct testing would be primary human bronchial epithelial (HBE) cells, their supply is limited and hard to come by [12]. The cell line BEAS-2B, an immortalised HBE cell line, is recognised as a valid cell type to screen chemical and biological agents to monitor cell differentiation [13,14,15]. Due to the similarities in gene expression of BEAS-2B to primary cultured cells, its use has currently expanded from testing of pollutants [16] and nanomaterials [17,18] to monitoring proinflammatory cytokine landscapes following infection [19]. 

In addition to cell models, modifications to in vitro culture mediums are now being developed to mimic host-specific nutrient profiles to uncover microbial pathogenicity [20]. The CF microenvironment is defined by excessive mucus, and to mimic this environment, we have formulated a modified artificial sputum medium (ASMDM-1) based on optimisation and refinement of the methods of Kirchner et al. (2012) [21], Sriramulu et al. (2010) [22], and Fung et al. (2010) [23]. This medium can approximate CF sputum in terms of concentrations of components and its physical properties. Previous studies have mainly used the medium to assess changes in the gene expression of *P. aeruginosa* chronic infections [23,24]. In a study by Wijers et al. (2016), a unique methodology was outlined where artificial sputum and A549 cells were used together to investigate *Burkholderia cenocepacia* infections in vitro [25]. However, their study did not assess the impact of a treatment of any kind following bacterial infection and focused exclusively on the use of alveolar cells. However, the literature has highlighted that CF opportunist species may also infect bronchial cells [26] and human macrophages [27] and bacterial colonisation may persist. CF epithelial cells are well known for their inability to mount an appropriate response to infection, allowing bacterial colonisation [28]. Lung macrophages are also a site of *A. xylosoxidans* infection [27]. Macrophages are sentinels of innate immunity with their main effector function being phagocytosis of bacterial cells [29]. While the literature has yet to confirm the site of infection of *A. xylosoxidans* in the CF lung, we hypothesised that it would have similar patterns of infection in HBE cells (BEAS-2B) and macrophages (THP-1). 

In this study, we aim to build on the use of artificial sputum and cell lines by substituting a human bronchial cell line (BEAS-2B) and differentiated monocytes (THP-1) to investigate the effect of a CT on biofilms of the opportunistic CF species: *A. xylosoxidans*. Ultimately, our objective is to create a reproducible model for CF infection by using artificial sputum medium and BEAS-2B cells to test bacterial infection and treatment toxicity. 

## 2. Materials and Methods

### 2.1. Preparation of Artificial Sputum Media (ASMDM-1)

Modified artificial sputum medium (ASMDM-1) was prepared using previously detailed methods [21,22,23]. ASMDM-1 contains mucin from porcine stomach (Sigma-Aldrich, Sydney, Australia), DNA from salmon testes (Sigma-Aldrich, Sydney, Australia), the iron chelator diethylene triamine penta-acetic acid (DTPA) (Sigma-Aldrich, Sydney, Australia), “10× Salt solution” made using NaCl (Sigma-Aldrich, Sydney, Australia), KCl (Sigma-Aldrich, Sydney, Australia), CaCl_2_ (Sigma-Aldrich, Sydney, Australia), 5× Bovine Serum Albumin (Sigma-Aldrich, Sydney, Australia), 10× Casamino acids (Bacto, Sydney, Australia), and egg yolk emulsion (Oxoid, Sydney, Australia) at concentrations found in an average CF patient [30,31]. The components were homogenised in sterile dH_2_O and filter sterilised overnight at 4 °C. Media was warmed to room temperature prior to use. To create a 20% ASMDM-1 solution, 20% of pure ASMDM-1 was added to 80% Dulbecco’s Modified Eagle Medium (DMEM) (Sigma-Aldrich, Sydney, Australia) or 80% RPMI-1640 (RPMI) (Sigma-Aldrich, Sydney, Australia) and filter-sterilised through a 0.22 μm membrane filter.

### 2.2. A. xylosoxidans Strains and Growth Conditions

*A. xylosoxidans* ATCC 27061™ (American Type Culture Collection, Manassas, VA, USA) and one representative CF strain of *A. xylosoxidans* (referred to as “AX 4365”) were used in this study (Table 1). CF strains isolated and identified using MALDI-TOF (Bruker, MA, United States) from patient sputum cultures (CF clinic, Royal Prince Alfred Hospital, Sydney, Australia) were stored at −80 °C. A CF specialist at the clinic collected sputum from patients and isolated this clinical strain of *A. xylosoxidans*. 

*A. xylosoxidans* isolates were grown in full strength in Tryptone soya broth (TSB, Oxoid, Sydney, Australia) and on Tryptone soya agar (TSA, ThermoFisher Scientific, Sydney, Australia) plates for 48 h (37 °C and 150 rpm). After 48 h growth, bacterial cells were pelleted by centrifugation (15 min, 1100 g, 20–24 °C) and washed once with 1 × phosphate buffer saline (PBS: 137 mM NaCl, 2.7 mM KCl and 10 mM phosphate, pH 7.41) (POCD Healthcare, Sydney, Australia) and centrifuged again for 10 min as above, before being suspended in the relevant cell media below. “Suspended bacterial cultures” referred to herein were maintained at OD_600nm_ 0.1 ± 0.02. 

### 2.3. Bronchial and Differentiated Monocyte Cell Lines and Propagation Conditions

BEAS-2B cells were maintained in 1 mL aliquots at −80 °C. Cells were resuscitated and cultured using DMEM supplemented with 10% (*v*/*v*) foetal bovine serum (FBS) (ThermoFisher Scientific, Sydney, Australia), penicillin (100 IU/mL), and streptomycin (100 μg/mL) (Sigma-Aldrich, Sydney, Australia) (referred to as “PenStrep”). Cells were grown to confluence in T25 cell culture flasks (Corning, New York, NY, USA) at 37 °C and 5% (*v*/*v*) CO_2_. To subculture, cells were harvested at 90% confluency using 0.12% *v*/*v* trypsin-EDTA (with phenol red) (Sigma-Aldrich, Sydney, Australia). Cells were collected in conical 50 mL falcon tubes and centrifuged (7 min, 125 g, 20 °C). The cell pellet was resuspended in supplemented DMEM or diluted ASMDM-1 for further experiments. 

THP-1 cells were maintained as above with the following exceptions. Cells were resuscitated and cultured in suspension using RPMI media, instead of DMEM, and supplemented with 10% *v*/*v* FBS and PenStrep. THP-1 suspension monocytes were differentiated into adherent macrophages by seeding at a density of 2 × 10^5^ cells/mL in a final concentration of 160 nM of phorbol-12-myristate-13-acetate (PMA) (Sigma-Aldrich, Sydney, Australia) for 48 h. Cells were washed with fresh RPMI media before use in experiments. 

### 2.4. Preparation of Treatments and Media formulation

CT formulations were derived from a previous publication [6] and are detailed in Table 2. The pH-adjusted NAC, NAC_neutral_, was used for the following experiments. NAC_neutral_, ciprofloxacin, and colistin stock solutions were prepared immediately before use by dissolving the respective powder in sterilised solutions of ASMDM-1 diluted in DMEM or RPMI, and in the case of ciprofloxacin, to a final concentration of 0.1 M HCl. All experiments were performed in 20% ASMDM-1. 

To prepare fresh NAC_neutral_ stock solutions (97.9 mg/mL) each time before experiments, NAC powder (Sigma-Aldrich, Sydney, Australia) was dissolved in sterile DMEM, and the pH was adjusted to 6.5–7.4 with 0.1 M NaOH (Appendix A). The antibiotic ciprofloxacin and colistin (Sigma-Aldrich, Sydney, Australia) stock solutions (900 μg/mL) were also prepared fresh before each experiment. Importantly, ciprofloxacin was specifically required to be dissolved in 0.1 M HCl which was created in DMEM. Both NAC and antibiotic solutions were filtered through a 0.22 μm membrane filter. 

### 2.5. Optimal Tolerance of Bronchial Epithelial Cells (BEAS-2B) in ASMDM-1

The optimal concentration of ASMDM-1 in BEAS-2B was determined using previously detailed methods [25] with modifications. Briefly, BEAS-2B cells were seeded in 24-well plates (Corning Corp, New York, NY, USA) at 5 × 10^4^ cells/mL. The plates were incubated at 37 °C (5% *v*/*v* CO_2_) until confluent. Following incubation, the media was discarded, and cells were washed once with 1 × PBS. DMEM or diluted ASMDM-1 was then added to the cells at varying concentrations; 0-, 20-, 40-, 60-, and 80% or 100% ASMDM-1. Following incubation, the supernatant was removed from all wells and plates and washed with 1 × PBS. 

To assess cell viability, cells were detached from plates by washing with 1 × trypsin-EDTA at 37 °C for 10 min. Cell density and viability (% live cells) was measured using trypan blue staining using a hemocytometer (Neubauer, Stallikon, Switzerland). To assess cell metabolic viability, 0.05% *w*/*v* resazurin solution (Sigma-Aldrich, Sydney) was added to adherent cells, and incubated further at 37 °C with orbital shaking (150 RPM). After 24 h, the fluorescence intensity of the BEAS-2B cells were determined at Ex_544nm_ and Em_590nm_ using the Tecan infinite M1000 pro plate reader (Tecan, Grödig, Austria). 

### 2.6. Cytotoxicity and Re-Growth of BEAS-2B and THP-1 following Treatment with Single and Combination Therapy

To determine the cytotoxicity of treatments on cells, the resazurin reduction assay was performed using metabolic viability as a surrogate for overall cell viability. In brief, BEAS-2B cells were seeded in 24-well plates to a density of 5 × 10^4^ cells/mL and incubated at 37 °C until confluence was achieved. Cells were then washed with 1 × PBS and treated with NAC_neutral_ in 20% ASMDM-1 mixed with DMEM. Treatments were removed at 1-, 3-, 6-, 12-, and 24 h and plates were washed twice with 1 × PBS. Treatments in 24-well plates were replaced with 0.05% w/v resazurin solution (Sigma-Aldrich, Australia) and after 6 h the fluorescence intensity of the BEAS-2B cells were determined at Ex_544nm_ and Em_590nm_ using the Tecan infinite M1000 pro plate reader. 

Regrowth was performed on BEAS-2B cells only, as described in the cytotoxicity assay as above, with the following exceptions. Treatments made in 20% ASMDM-1 were added to the wells for 3 h. Following the removal of treatments, the media was replaced with the relevant 20% ASMDM-1 and cells were incubated for a further 24 h before the resazurin viability stain was added and viability quantified as above. 

Following on from results of the above experiments, cytotoxicity following 3 h treatment was assessed in differentiated THP-1 cells. Cells were seeded, differentiated, and then washed twice with RPMI media to remove remaining PMA and treatments made in 20% ASMDM-1 and 80% RPMI medium were added for 3 h. After treatment, media was removed and replaced with 0.05% *w*/*v* resazurin solution (Sigma-Aldrich, Sydney, Australia) and quantified as above. 

### 2.7. Bacterial Mucin Adhesion Assay

Adhesion of *A. xylosoxidans* strains to mucin matrices was measured according to the modified methods [32,33]. In brief, 96-well microtiter plates were coated with 30 mg/mL porcine stomach mucin type IV (Sigma-Aldrich, Sydney, Australia) (made in 50 mM carbonate buffer) for 24 h at 4 °C. Control wells were carbonate buffer alone and were replaced with 1% (*v*/*v*) Tween20 × PBS to saturate uncoated binding sites for 1 h at room temperature. Wells were subsequently washed with 1 × PBS before 200 μL of diluted bacterial suspensions in 1 × PBS were added for 48 h (37 °C and 150 RPM) to allow for mature biofilm formation. Incubated alongside this, control wells with the same bacterial suspension were included without any mucin. After incubation, wells were washed in 0.05% Tween20 × PBS to remove any non-adherent cells before CT (detailed in Table 2) and were added for 3 h (37 °C and 150 RPM). Wells were then washed twice in 1 × PBS and heat fixed for 1 h in a 65 °C dry oven. Adhered cells were stained with 0.1 mg/mL crystal violet overnight (37 °C and 150 RPM) and washed twice with 1 × PBS to remove any excess stain. Excess stain was dissolved by adding 50 mM citrate buffer to all wells for 1 h and reading absorbance at 595 nm using a plate reader. 

### 2.8. Bacterial Invasion Assay of Bronchial and Monocyte-Derived Macrophage Cells 

BEAS-2B and THP-1 cells were seeded at 5 × 10^4^ cells/mL or 2 × 10^5^, respectively, in 24-well plates. BEAS-2B cells were grown until confluent, while THP-1 cells were differentiated in PMA for 48 h prior to use. Cells were washed with 1 × PBS twice. Diluted bacterial cultures were then added to the BEAS-2B cells at a multiplicity of infection (MOI) of ~1 in 10 for 1 h. Following bacterial incubation, the cells were washed once with 1 × PBS. Gentamicin was added to wash the cells 3 times for a total of 30 min to kill any extracellular bacteria. Cells were then treated with CT (outlined in Table 2) for 3 h. Following treatment, cells were then washed twice with 1 × PBS and treated with Triton X-100 (0.4% *vol*/*vol*) for 20 min to lyse epithelial cells and release intracellular bacteria. A Whitley automatic spiral plater (WASP) was used (Don Whitley Scientific, Bingley, West Yorkshire, UK) to establish a colony-forming unit (CFU/mL) count. The WASP automatically plated 50 μL of serially diluted bacterial suspension on TSA plates. Plates where incubated for 48 h at 37 °C in a static incubator, following which the plate colonies were enumerated and expressed as CFU/mL. 

### 2.9. Statistical Data Analysis

Statistical significance difference was analysed using GraphPad Prism version 9.0 (San Diego, CA, USA). Data were analysed using unpaired Student’s *t*-tests (with Welch’s correction) and multiple comparison tests (Kruskal–Wallis test with Dunn’s correction). The difference was considered statistically significant if cut-offs were *p* > 0.05 (n), *p* ≤ 0.05 (*). All experiments were conducted at least in n = 3 biological replicates.

## 3. Results

### 3.1. BEAS-2B Cell Viability Decreases with Increasing Concentrations of ASMDM-1 

ASMDM has been used to simulate the presence of mucus in the CF lung [23,24,34]; specifically the modified version, ASMDM-1, as used in this study. Cell viability and density quantification were used as supplementary methods to determine the concentration of ASMDM-1 that could be tolerated by BEAS-2B cells (Figure 1). All statistical comparisons were performed against this untreated (0% ASMDM-1) control. Untreated cells had 100% viability evidenced by 100% metabolic viability and a 1.8 × 10^6^ cells/mL live cell density. These results were mirrored in Figure 1b, where cell density began to reduce significantly from 40% ASMDM-1 or higher. There was no significant difference between 20% ASMDM-1 and 0% live cell control. Therefore, the following experiments were performed in 20% ASMDM-1 (80% DMEM) media. 

### 3.2. BEAS-2B Cells Regrow following Three-Hour Treatment Time

The duration that the BEAS-2B cells in 20% ASMDM-1 could withstand and regrow following treatment challenge was assessed by determining resazurin metabolic viability over time (Figure 2). As the cytotoxicity of the antioxidant component, NAC_neutral_, has not yet been determined in this model, the results in Figure 2a highlight the cytotoxicity of all individual NAC_neutral_ concentrations (1-, 2-, 4.1-, and 8.2 mg/mL over 24 h). The cells experienced an initial decrease in viability at the one-hour time point but were able to recover at the three-hour time point and eventually maintained nearly 100% viability for 1- and 2 mg/mL concentrations. Conversely, viability dropped to 56.4% and 48.3% for 4.1- and 8.2 mg/mL NAC_neutral_, respectively. 

The three-hour time point was chosen in order to observe the ability of cells to restore viability for all treatment options, including NAC_neutral_ treatments (Figure 2b). Following a three-hour treatment challenge, represented by red bars, viability was measured after 24 h, represented by yellow bars. Cells subjected to CT did not experience significant decreases in viability both in the initial challenge and in the regrowth phase, with the exception of combinations G and J, both colistin based CTs. Combinations G and J both significantly decreased in viability after three hours to 82.9% and 44.3%, respectively. However, both conditions recovered to 100% viability following regrowth after 24 h. 

There was a significant decrease in viability for most conditions, especially for all individual antibiotic treatments (*p* ≤ 0.05). However, cells were able to recover viability to 100% following 24 h in 20% ASMDM-1 growth medium. There were some exceptions to this, such as for 64 μg/mL ciprofloxacin, where viability significantly decreased to 27.5% at three hours and recovered to 85% after 24 h. Similarly, both 8- and 16 μg/mL colistin treatments decreased cell viability to 39.1% and 32.9% but this recovered to 70% for both conditions after 24 h (Figure 2b). 

### 3.3. THP-1 Cells Maintain Metabolic Viability after Three Hours of Combination Treatment

THP-1 cells were subjected to a three-hour treatment challenge with CT and individual components to assess metabolic viability against an untreated control (Figure 3). Application of treatments on differentiated THP-1 cells resulted in a significant reduction in metabolic viability, specifically for individual antibiotic treatments with average reductions of 65% compared to the untreated control (*p* < 0.05). Similarly, significant reductions (*p* < 0.05) ranging from 11–20% were observed in NAC_neutral_ individual treatments compared with the untreated control. However, when both components were applied in combination, all but one combination, F, did not experience any significant reductions in viability after three hours (*p* > 0.05) and remained metabolically viable compared to the untreated control. 

### 3.4. Mucin Provides a Suitable Substrate for Bacterial Adhesion and Biofilm Formation

Adhesion of *A. xylosoxidans* to mucin and its subsequent response to a three-hour treatment challenge was measured relative to control non-coated wells and has been expressed as % bacterial biomass in Figure 4. 

In Figure 4a, significant reductions in biomass were observed when using colistin-based CT, with an average percentage biomass reduction of 65% from the normalised untreated control. These reductions were also significant in comparison to the individual components of the CT, with an average percentage biomass reduction of 20% recorded in both NAC_neutral_ and colistin components (*p* ≤ 0.05). 

The representative clinical strain AX 4365 as shown in Figure 4b had fewer significant reductions compared with its ATCC counterpart. However, all significant reductions in biomass were identified with colistin combinations; specifically, E, F, and I. Average percentage biomass reductions of 70% were recorded compared with the normalised, untreated controls. When individual components of CT were compared, average biomass reductions of 7% were recorded in NAC_neutral_ components and 18% for colistin components (*p* ≤ 0.05).

### 3.5. A. xylosoxidans Intracellular Bacterial Load Is Not Universally Reduced When Assessed in BEAS-2B + ASMDM-1 Cell Model

Similar to Section 3.6, BEAS-2B cells were infected with *A. xylosoxidans* strains, and following a three-hour treatment exposure, bacterial loads were also significantly reduced predominately in colistin-based CT (Figure 5). Bacterial load reductions were compared against the untreated control and both individual components of the combination to determine statistical significance. 

In Figure 5a, ATCC 27061 showed greater significant reductions when using all 16 μg/mL colistin + NAC_neutral_ CT, as seen in combinations H, I, and J. All combinations had a ~1.5 log_10_ reduction compared with untreated controls (*p* ≤ 0.05) and also displayed synergy during planktonic synergy testing. The other combination, E, was also a colistin-based CT and displayed synergy during planktonic testing and had similar ~1.5 log_10_ reductions as other combinations. 

In Figure 5b, strain AX 4365 did not show significant reductions in comparison to Figure 5a with only one combination leading to significant reduction (D). This was a 64 μg/mL ciprofloxacin combination which originally displayed indifference during checkerboard synergy planktonic testing. However, its reduction was significant (*p* ≤ 0.05) with a ~2 log_10_ reduction compared with the untreated control. 

### 3.6. Synergistic Colistin Combination Therapy Reduced A. xylosoxidans Invasion in THP-1 Cells

To determine intracellular invasion in THP-1 macrophages, differentiated cells were infected with *A. xylosoxidans* for one hour followed by a three-hour treatment exposure, where bacterial loads were significantly reduced, predominantly in colistin-based CT (Figure 6). 

ATCC 27061 successfully invaded the THP-1 cells and intracellular bacterial loads were reduced compared with both untreated control and individual components of the treatments D, H, and I (Figure 6a). Combination J was additive in planktonic states but did not display a significant intracellular bacteria reduction when compared to 16 μg/mL colistin alone (*p* > 0.05). Combinations H and I both displayed synergy in planktonic states which was translated into significant reductions of ~2 log_10_ compared with the untreated control in this cell + (ASMDM-1) model. Combination I, while considered indifferent under planktonic conditions, showed significant reductions of ~0.5 log_10_ compared with the untreated control (*p* ≤ 0.05). 

Strain AX 4365 showed reductions similar to its ATCC counterpart (Figure 6b). The CFU/mL reductions were mirrored here for combinations H and I, with two ciprofloxacin combinations, B and C, also resulting in significant reductions. The colistin-based combinations displayed synergy in planktonic cultures also resulting in significant ~2 log_10_ reductions compared with the untreated control. Combinations B and C initially displayed indifference in planktonic cultures; however, their ~2 log_10_ reductions compared with the untreated control were considerably significant despite the initially displayed indifference (*p* ≤ 0.05). 

## 4. Discussion

The goal of this study was to optimise an in vitro cell culture plus ASMDM-1 model as a surrogate to assess the efficacy of antioxidant-based CT on *A. xylosoxidans* infection. A hallmark of the CF microenvironment is the accumulation of mucus [35]. Therefore, to mimic this environment [32], we substituted general growth media for diluted ASMDM-1 [23]. As such, this study allowed for a more accurate assessment of CF by bringing together two previous studies [6,25]. This study assesses the impact of synergistic and additive CT in an ASMDM-1 model using bronchial cells and lung macrophages to model *A. xylosoxidans* infection. We have separated this study into three distinct stages. Firstly, optimisation of the bronchial/bacterial cell to sputum ratio. Secondly, testing cytotoxicity of treatments on cells alone in the optimised sputum model, and, thirdly, assessment of CT in the context of bacterial infection within the live cell/sputum model. While the site of infection of *A. xylosoxidans* in the lungs is yet to be confirmed in either bronchial epithelial cells or macrophages, we have chosen these two cell populations to separately model infection and treatment. To avoid introducing too many external variables in the form of diverse clinical strains, only two *A. xylosoxidans* strains were tested. 

Our model was optimised based on the concentration of ASMDM-1 that could be tolerated by both BEAS-2B cells and bacteria. As a result, 20% ASMDM-1 was chosen to maintain cells as it showed no significant differences in viability between the conventional growth medium control in cells alone (Figure 1) (Appendix A), bacterial growth alone (Appendix A), and did not hinder imaging capabilities (Appendix A). These factors formed the cornerstone of the sputum model as it allowed us to monitor treatment cytotoxicity in viable cells in subsequent experiments. 

We then wanted to optimise the treatment time with CT by determining its cytotoxicity within the lung cell/ASMDM-1 model on both BEAS-2B and THP-1 cells without bacteria. Unlike a biofilm-only in vitro model, live cells pose a unique challenge in that most treatments would not likely be tolerated for extensive periods of time. As such, the “treatment time” was defined as the threshold where cells may experience cytotoxicity but still could maintain viability to regrow when the treatment was replaced with growth medium (Figure 2). Cytotoxicity testing is a requirement of most regulatory authorities. The appropriate in vitro methods have been reviewed by Liu et al. [36], who stipulate that any new antimicrobials must be tested in a live cell model for a period of 24 h. As ciprofloxacin and colistin are already regularly used clinically [37], we chose NAC_neutral_ as our signpost for cytotoxicity as its use at this neutralised pH has not yet been approved for inhaled human therapy [38]. Our results showed that BEAS-2B cells tolerated NAC_neutral_ well over a time course of 24 h (Figure 2a). However, we chose to further investigate regrowth of cells following the three hours of combination treatment as the BEAS-2B maintained cell viability close to or above 50% using individual treatments. Following the three-hour treatment challenge, all conditions were able to recover to 100% viability in both BEAS-2B (Figure 2b) and THP-1 cells (Figure 3), even in the cases of mildly acidic pH ciprofloxacin CT. This result confirmed that within the 20% ASMDM-1/lung cell model, cells could retain their viability following a three-hour treatment time. This optimised cell and treatment model allows us to monitor *A. xylosoxidans* intracellular infections in subsequent experiments. 

We then confirmed the ability of *A. xylosoxidans* to adhere and colonise to the mucin component in ASMDM-1 and the impact of the three-hour treatment on the resultant biomass (Figure 4). We used Type IV porcine stomach mucin-coated plates, the structure of which is similar to the human mucin molecule, and MUC5AC [39] to provide a substrate for *A. xylosoxidans* biomass formation. Following the three-hour treatment time, there was a significant reduction in biomass using colistin-based CT for all strains (Figure 4a,b), which suggested that bacteria may colonise in the sputum layer of the model and the CT are able to reduce biomass. 

Finally, we tested the effect of CT on *A. xylosoxidans* infection within the cell/sputum model in BEAS-2B and THP-1 cells (Figure 5 and Figure 6). Our research question was: did the CT conditions significantly lower bacterial loads in comparison to the untreated control and both individual components of the CT? Significance was only displayed in those instances. We compared any significant combinations from this study with previously published data [6], assigning relevant synergy classifications. In THP-1 cells, there were significant and synergistic reductions in ATCC 27061 and AX 4365 bacterial loads when the colistin-based combinations, H and I, were used (Figure 6). In BEAS-2B cells, while the combinations H, I, and J, were able to reduce ATCC 27061 bacterial loads, this was not mirrored in AX 4365 where only one ciprofloxacin-based combination, D, could reduce bacterial loads (Figure 5). Thus, while colistin-based CT may provide adequate reductions in some cases, it does not address all *A. xylosoxidans* infections in the CF lung due to strain variability. 

The main benefit of conducting work in a live cell model versus a flow cell or microplate is that the cells provide a live medium with changing bacterial dynamics. ASMDM-1 was used to create a replicable model and provides the benefit of ease of access and consistency due to defined concentrations in the formulation. These two features are not present if using an extracted sputum sample, which is complicated by patient-to-patient variation [34,40]. 

*A. xylosoxidans* has received less attention than other CF pathogens, but is emerging as an opportunist, causing progressive lung dysfunction in chronic infection [41]. The cell/sputum model could provide greater insights into the *A. xylosoxidans* genes affected during colonisation and treatment challenge. In a study by Jakobsen et al. (2013), the entire genome of pathogenic *A. xylosoxidans* NH44784-1996 was studied [42]. The *A. xylosoxidans* genome contained the *pgaABCD* operon. This encodes the polysaccharide β-1,6-GlcNAc, involved in cell-to-cell adherence and cell surface adherence, and contributed to biofilm formation [43] within the cell-sputum model. 

One limitation is the chosen ASMDM-1 concentration of 20%. In the CF lung, the defective CFTR ion transport results in characteristic static mucus accumulation which eventually progresses to bronchiectasis [44]. In our in vitro study, however, we required viable cells in which to study bacterial infection, which meant using “suffocating” sputum concentrations was not possible. One could rectify this by conducting optimisation using gas-permeable wells to regulate oxygen bioavailability to cells [25,45]. However, this poses the same issue of misrepresentation of the CF lung where the mucus created varied O_2_ gradients [46] and patients generally report chronic hypoxemia [47,48]. As such, in vitro models will always have their limitations, but without them it is nearly impossible to derive any inkling of infection capacity or treatment efficacy.

Future work could assess other emergent bacterial species, such as *Stenotrophomonas maltophilia* [49], and improving the model in which testing is performed. *A. xylosoxidans* mucin adhesion suggests an investigation into relative bacterial colonisation in the ASMDM-1 layer versus the adherent cell layer is warranted. This could be accomplished by fluorescently tagging the bacteria and performing live cell confocal microscopy to monitor colonisation and eventual biofilm formation [4,50,51]. Another future direction could be the co-culture of THP-1 and BEAS-2B cells to investigate the cell-to-cell signalling and proinflammatory [19] responses to bacterial infection, thus shedding light on how CT either mitigates or abrogates cell processes. Studies by Li et al. (2012) assessed the effect of asbestos particles in transwells on the proinflammatory responses of THP-1 cells in the apical chamber and their effects on bronchial epithelial cells in the basolateral chamber [19]. This model may be altered to suit specific requirements and could highlight the impact of cytokine responses and oxidative stress. 

## 5. Conclusions

In conclusion, our study has optimised a cell/artificial sputum model using two relevant lung cell populations, BEAS-2B and THP-1, to study the effects of CT on *A. xylosoxidans* infection. We found that colistin-based CT is well tolerated by both cell types and may significantly reduce bacterial loads in vitro. This pilot study provides a foundation to study other bacterial opportunists that infect the CF lung to observe infection and for recording CT effectiveness. This model can also act as a springboard to more complex co-culture models and eventually more streamlined, optimised, and budget-friendly testing in air–liquid interface cell models [52]. 

## Figures and Tables

**Figure 1 biomedicines-10-02886-f001:**
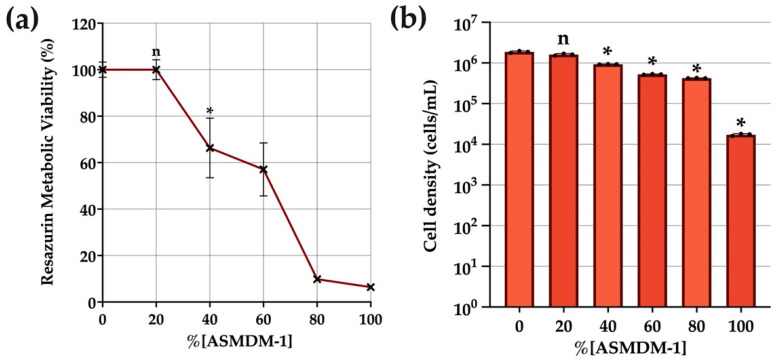
The effect of ASMDM-1 on BEAS-2B cell density and viability. BEAS-2B cells were grown to confluence in 24-well plates and exposed to varying concentrations of ASMDM-1 diluted in DMEM media. After 24 h incubation, the (**a**) cell viability (resazurin metabolic viability %) and (**b**) cell density (cells/mL) were compared with a normalised, untreated control. Statistical analyses were performed using unpaired Student’s *t*-tests (with Welch’s correction). Significance cut-offs were *p* > 0.05 (n), *p* ≤ 0.05 (*). Data represent an average of n = 3 biological replicates.

**Figure 2 biomedicines-10-02886-f002:**
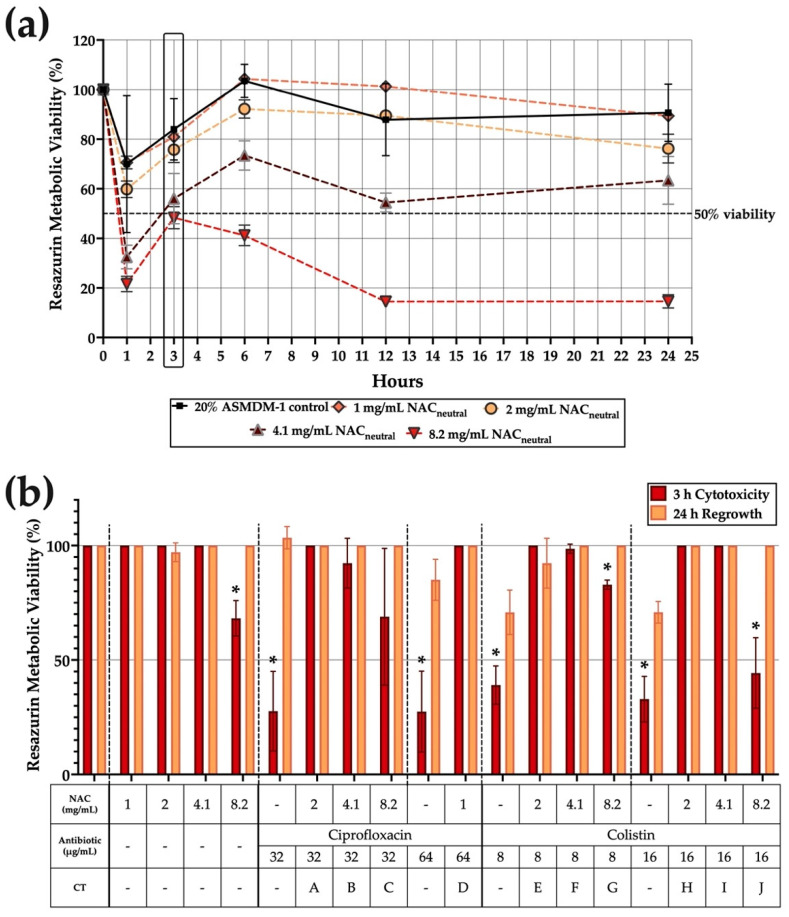
BEAS-2B cells experience minimal cytotoxicity following three-hour combination treatment challenge. BEAS-2B cells were grown to confluence and the cytotoxicity of NAC_neutral_ CT made in 20% ASMDM-1 (as detailed in Table 1) was tested. (**a**) Cytotoxicity of NAC_neutral_ was measured using resazurin metabolic viability staining over 24 h to determine viability. (**b**) Regrowth following three hours of treatment with CT and individual components were assessed after 24 h. (*) indicates the results are statistically significant (*p* ≤ 0.05) based on unpaired Student’s *t*-tests (with Welch’s correction).

**Figure 3 biomedicines-10-02886-f003:**
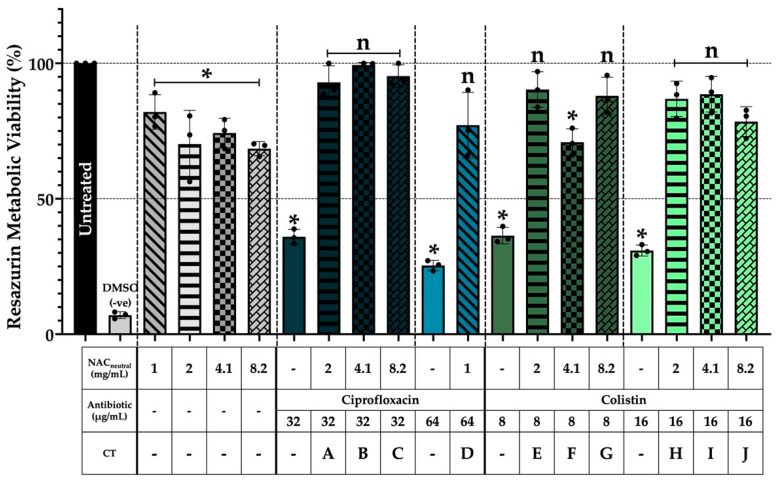
THP-1 cells maintained metabolic viability after three-hour combination therapy treatment challenge. THP-1 cells were grown in suspension until a density of 2–4 × 10^5^ cells/mL was achieved. Cells were differentiated by seeding into 24-well plates at a density of 2 × 10^5^ cells/mL in a final concentration of 160 nM PMA for 48 h. Cells were washed and combinations were added to a concentration of 20% ASMDM-1 and 80% RPMI medium for three hours and metabolic viability was measured using resazurin staining. (*) indicates the results are statistically significant (*p* ≤ 0.05) based on unpaired Student’s *t*-tests (with Welch’s correction). Experiments were performed in biological triplicate (n = 3).

**Figure 4 biomedicines-10-02886-f004:**
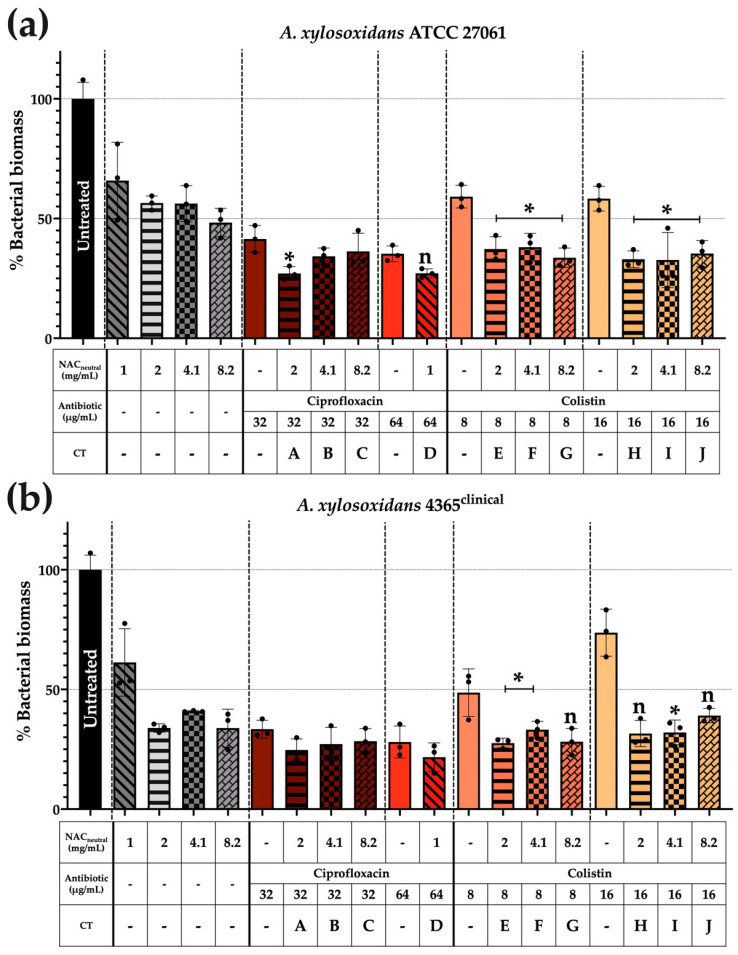
Mucin substrate-enhances *A. xylosoxidans* bacterial adhesion and reduced biomass occurred after a three-hour treatment challenge. Porcine mucin (Type IV)-coated plates were used to monitor bacterial adhesion and subsequent biofilm formation. Briefly, diluted bacterial cultures were added to mucin-coated wells at an OD_600_ 0.1 ± 0.02 for 48 h. Wells were washed with 1 × PBS. Treatments prepared in 1 × PBS were added for three hours. Wells were washed three times before heat fixing in a dry oven at 65 °C for one hour. Wells were stained with 0.1% (*w*/*v*) crystal violet for 24 h and de-stained with citrate buffer for one hour prior to absorbance being read at 595 nm. Biomass was quantified against a normalised, untreated control and displayed as “% biofilm biomass”. Strains shown are (**a**) ATCC 27061 and (**b**) AX 4365. (*) indicates the results are statistically significant (*p* ≤ 0.05) (*) based on unpaired Student’s *t*-tests (with Welch’s correction). Significance was only displayed if CT was significant compared with untreated controls and both individual components of CT. Experiments were performed in biological triplicate (n = 3).

**Figure 5 biomedicines-10-02886-f005:**
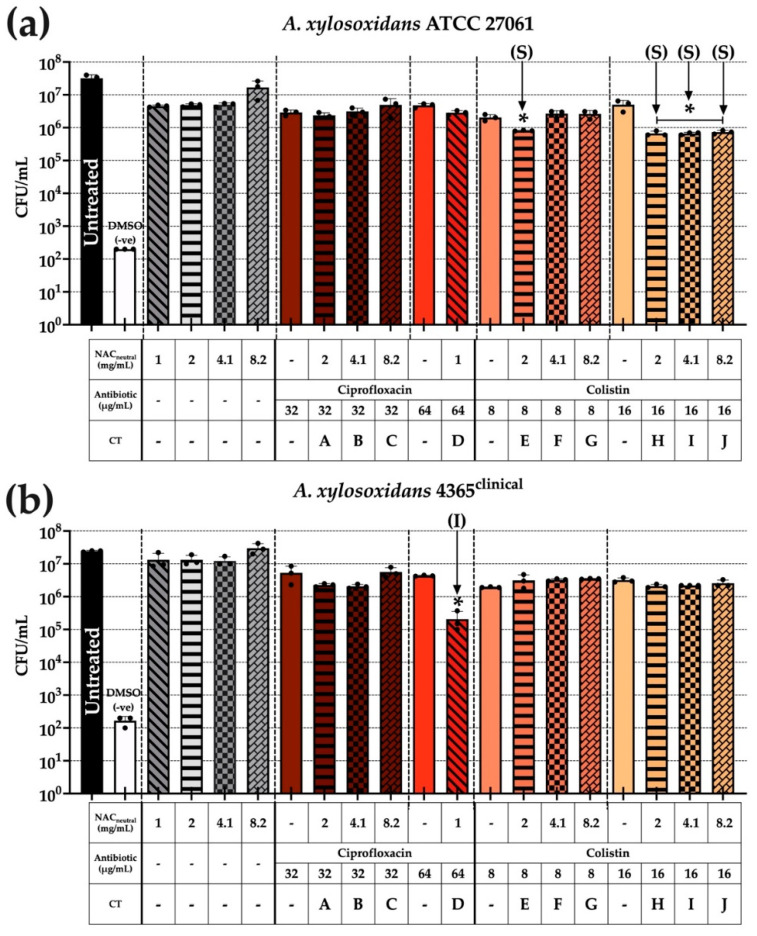
Combination treatment challenge against *A. xylosoxidans* invasion in BEAS-2B + 20% ASMDM-1 cell model. Confluent BEAS-2B cells were washed and infected with OD_600_ 0.1 ± 0.02 for one hour to allow for intracellular invasion. Following this, cells were washed thrice with 100 μg/mL gentamicin before addition of combinations in a final concentration of 20% ASMDM-1 and 80% DMEM medium for three hours. Cells were washed twice with 1 × PBS before being lysed and intracellular bacterial loads were enumerated using CFU/mL. (**a**) ATCC 27061 and (**b**) AX 4365. Correlation with planktonic synergy was determined previously and is denoted as (**S**) synergy and (**I**) indifferent. (*) indicates the results are statistically significant (*p* ≤ 0.05) (*) based on unpaired Student’s *t*-tests (with Welch’s correction). Experiments were performed in biological triplicate (n = 3).

**Figure 6 biomedicines-10-02886-f006:**
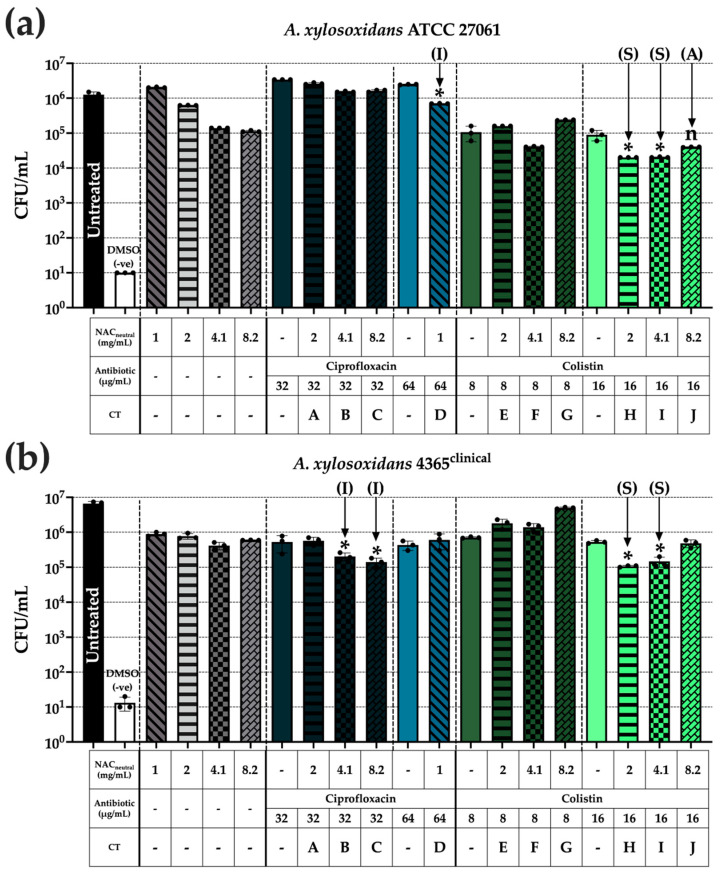
THP-1 cell invasion by *A. xylosoxidans* may be reduced with synergistic combinations of colistin. PMA-differentiated THP-1 cells were washed and infected with diluted bacterial cultures for one hour to allow for intracellular invasion. Following this, cells were washed thrice with 100 μg/mL gentamicin before adding CT [6] in a final concentration of 20% ASMDM-1 and 80% RPMI medium for three hours. Cells were washed twice with 1 × PBS before being lysed and intracellular bacterial loads were enumerated using CFU/mL. Strains shown are (**a**) ATCC 27061 and (**b**) AX 4365. Correlation with planktonic synergy has been determined previously [6] and is denoted on relevant data as follows: (**S**), synergy, (**A**) additive, and (**I**) indifferent. (*) indicates the results are statistically significant (*p* ≤ 0.05 (*) or not significant (*p* > 0.05) (n) based on unpaired Student’s *t*-tests (with Welch’s correction). Experiments were performed in biological triplicate (n = 3).

**Table 1 biomedicines-10-02886-t001:** Source and antibiotic profiles of *A. xylosoxidans* strains used in this study.

Strain	Source	Antibiotic Profile				
		TZP110	CAZ30	MEM10	CIP5	SXT25
ATCC 27061 ^TM^	ATCC (Manassas, Virginia) (Ear Discharge)	S	I	S	R	R
AX 4365 ^+^	CF Sputum	NA	NA	NA	NA	R

^+^ Clinical isolate of *A. xylosoxidans* was collecetd and de-identified at the respiratory clinic, Royal Prince Alfred Hospital (RPAH), Sydney, Australia, prior to use in this study. Antibiotic susceptibility tests were performed at the microbiology department, RPAH, using VITEK2 as per the manufacturer’s instructions, using CLSI breakpoints for “other non-enterobacterales” (1). For colistin no breakpoints were detected. Breakpoints for antibiotics are Piperacillin-Tazobactam (TZP100) = 110 μg/mL; Ceftazidime (CAZ30) = 30 µg/mL; Meropenem (MEM10) = 10 μg/mL; Ciprofloxacin (CIP5) = 5 μg/mL; Trimethoprim-Sulfamethoxazole (SXT25) = 1.25/23.75 μg/mL. S, I, and R represent breakpoints intermediate and resistance, respectively, whereas NA = not available.

**Table 2 biomedicines-10-02886-t002:** Combination therapy (CT) used in study was derived from Aiyer et al. (2021) [6] and was based on a synergy fractional inhibitory concentration index checkerboard assessment.

Letter Reference	Neutral NAC (mg/mL)	Antibiotic (μg/mL)
A	2	32 μg/mL Ciprofloxacin
B	4.1	
C	8.2	
D	1	64 μg/mL Ciprofloxacin
E	2	8 μg/mL Colistin
F	4.1	
G	8.2	
H	2	16 μg/mL Colistin
I	4.1	
J	8.2	

## Data Availability

Not applicable.

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
