# Peer review of "The Efficacy of an N-Acetylcysteine–Antibiotic Combination Therapy on Achromobacter xylosoxidans in a Cystic Fibrosis Sputum/Lung Cell Model"

_biomedicines, 2022, doi:10.3390/biomedicines10112886_

Round 1

Reviewer 1 Report

Review (Biomedicines), N-acetylcysteine-antibiotic combination therapy on Achromobacter xylosoxidans

The manuscript aims to optimize an in vitro culture model with added artificial sputum in order to evaluate combination therapy of bacteria in lungs of patients with cystic fibrosis. Two strains of A. xylosoxidans were selected, and different levels of NAC, colistin and ciprofloxacin were investigated. After clarification of appropriate medium, incubation conditions and time, the main test parameter was killing of intracellular bacteria.

The study is extensive and rather clearly presented. I have only one concern regarding the techniques, which relates to the identity of the A. xylosoxidans strains. What is ATCC 27601TM? The TM probably relates to the trademark of ATCC (?), but in the catalogue, 27601 refers to a strain of Streptomyces. PubMed search of Achromobacter strain Ax 4365 only identifies the former publication by the authors (Antibiotics 2021). Achromobacter is a saphrophytic bacterial genus that has attracted attention because of its association with chronic lung disease, and A. xylosoxidans is the type species of the genus. However, there are multiple species in this genus, and several of them – including xylosoxidans, insuavis, ruhlandii – are common in CF sputa. Currently, identification to species level cannot be accomplished by MALDI-TOF MS, or 16S rRNA gene sequencing, but is usually performed by nrdA sequencing. It was recently shown that substitutions in murein peptide ligase Mpl (conferring broad resistance) could easily be induced in Achromobacter ruhlandii, but not in other tested species of the genus. Although only minor differences were found between the two investigated strains, the authors should identify their strains to the species level.

Minor points.

 - The Discussion section could be shortened, because the relevant findings are clearly presented in the Results section

 - A. xylosoxidans is mostly written in italics, but not on page 3 (thrice) and p11. On p12, in the heading, the bacterium is underscored, perhaps to differentiate from the rest of the text in italics; usually bacterial names are written as standard text, to differentiate from letters in italics.

Author Response

Responses to Reviewer comments: Manuscript ID biomedicines-1886143

Reviewer #1: 

  1. I have only one concern regarding the techniques, which relates to the identity of the xylosoxidans strains. What is ATCC 27601TM? The TM probably relates to the trademark of ATCC (?), but in the catalogue, 27601 refers to a strain of Streptomyces. PubMed search of Achromobacter strain Ax 4365 only identifies the former publication by the authors (Antibiotics 2021). Achromobacter is a saphrophytic bacterial genus that has attracted attention because of its association with chronic lung disease, and A. xylosoxidans is the type species of the genus. However, there are multiple species in this genus, and several of them – including xylosoxidans, insuavis, ruhlandii – are common in CF sputa. Currently, identification to species level cannot be accomplished by MALDI-TOF MS, or 16S rRNA gene sequencing, but is usually performed by nrdA sequencing. It was recently shown that substitutions in murein peptide ligase Mpl (conferring broad resistance) could easily be induced in Achromobacter ruhlandii, but not in other tested species of the genus. Although only minor differences were found between the two investigated strains, the authors should identify their strains to the species level.

Response: Thank you for your comments. We have addressed its various components below:

With respect to the ATCC numbering, we sincerely apologise as we have incorrectly cited the ATCC number as “27601” when the number is meant to be “27061” (https://www.atcc.org/products/27061) This has been adjusted in the manuscript in all figures (Table 1 and Figure 5, 6, and 7 and Figure S3) and the following line numbers: 147, 371, 383, 402, 411, 435, 557 and 559.

With respect to the A. xylosoxidans clinical strain AX 4365 and its identification, this was all performed following isolation at the Royal Prince Alfred hospital (RPAH) microbiology department, Sydney, NSW, Australia. They use an updated database that is able to identify AX4365 strain down to the species level on the MALDI-TOF (Bruker). This strain was de-identified by the hospital following species identification and was gifted to the lab. In addition to the above MALDI-TOF ID database, there are recent studies that cite the use of updated databases when using MALDI-TOF which allows successful identification of Achromobacter species [1, 2]. To clarify the identification of AX 4365 to the species level prior to gifting to our lab we have added the following in the manuscript:

Line 147 “strains isolated and identified using MALDI-TOF (Bruker, Massachusetts, United States)

  1. The Discussion section could be shortened, because the relevant findings are clearly presented in the Results section

Response: Thank you for your comment. Discussion section has been shortened as required at the following lines:

Line 441 - 464:        

The goal of this study was to optimise an in vitro cell culture plus artificial sputum ASMDM-1 model as a surrogate to assess the efficacy of antioxidant-based CT on A. xylosoxidans infection. A hallmark of the CF microenvironment is accumulation of mucus [3]. Therefore, the use of a relevant growth media directly affects the appropriate assessment of bacterial infection and subsequent treatment. To the CF lung environment to mimic this environment [4], to model bacterial infection, we substituted general growth media for diluted ASMDM-1 [5]. As such, this study allowed for a more accurate assessment of CF by bringing together two previous studies [6, 7]. This study assesses the impact of synergistic and additive CT in an artificial sputum ASMDM-1 model using bronchial cells and lung macrophages to model A. xylosoxidans infection. We have separated this study into This study has three distinct stages. Firstly, optimisation of the bronchial/bacterial cell to sputum ratio. Secondly, testing cytotoxicity of treatments on cells alone in the optimised sputum model, and, thirdly, assessment of CT in the context of bacterial infection within the live cell/sputum model. While the site of infection of A. xylosoxidans in the lungs is yet to be confirmed in either bronchial epithelial cells or macrophages we have chosen these two cell populations to separately model infection and treatment. , with the former as the base for our model. To provide the sputum component, we used ASMDM-1; a medium traditionally used to study bacterial pathogenicity in vitro [23]. To avoid introducing too many external variables in the form of diverse clinical strains, only two A. xylosoxidans strains were tested.

 Line 470:

…These factors resulted in the successful maintainence of BEAS-2B cells was formed the cornerstone…

Line 481-485:

The appropriate in vitro methods have been reviewed by Liu et al [8]; which outlines stipulates that any new antimicrobials must be tested in a live cell model for a period of 24 h. As C ciprofloxacin and colistin are already regularly used clinically [9], . Therefore, we chose NACneutral as our signpost for cytotoxicity as its use at thisa neutralised pH has not yet been approved in the context offor inhaled human therapy [10]. Our results showed that BEAS-2B cells tolerated NACneutral….

Line 496- 502:

Before, we commenced intracellular infections, we wanted toWe then confirmed the ability of A. xylosoxidans to adhere and colonise to the mucin component in ASMDM-1 and the impact of 3 h treatment on the resultant biomass (Figure 4). We used Type IV porcine stomach mucin coated plates, the structure of which is like the human mucin molecule, MUC5AC [11] to provide a substrate for A. xylosoxidans biomass formation. Following the 3 h treatment time there was a significant reduction in biomass using colistin-based CT for all strains (Figure 4a and 4b) which suggested that while bacterial may colonise in the sputum layer of the model we have created and the CT are still able to reduce biomass.

Line 556 - 561:

…In THP-1 cells, Tthere were significant and synergistic reductions in ATCC 27061 and AX 4365 bacterial loads when the colistin-based combinations, H, and I, were used (Figure 6). In BEAS-2B cells, while the combinations H, I, and J, were able to reduce ATCC 27061 bacterial loads this was not mirrored in AX 4365 However, these results were not mirrored in BEAS-2B cells which displayed more variation. While the ATCC bacterial loads could be reduced using colistin0based combinations H, I, and J, the same results were not observed in AX 4365, where only one ciprofloxacin-based combination, D, could reduce bacterial loads (Figure 5). Therefore Thus, while colistin-based CT may provide adequate reductions in some cases, it is not a magic bullet to address all A. xylosoxidans infection in the CF lung due to natural strain variability.

Line 563 - 567:

By adding ASMDM-1, the nutritional components resemble the mucus availability in the CF lung [32]. ASMDM-1 used in the context of creating to create a replicable model provides the benefit of ease of access and consistency due to defined concentrations in the formulation; two features that are not present if using extracted sputum sample, which is complicated shows by patient-to-patient variation [12, 13].

Line 569 - 576:

  1. xylosoxidans has received less attention than other CF pathogens, but is emerging as an opportunist, as treatment is focused on other dominant species, causing progressive lung dysfunction in chronic infection [14]. The cell/sputum model could provide greater insights into the A. xylosoxidans genes affected during colonisation and treatment challenge and persistence during chronic infection. In a study by Jakobsen et al. (2013); the entire genome of pathogenic A. xylosoxidans NH44784-1996 was studied [15]. The A. xylosoxidans genome contained the pgaABCD operon, which has the capacity to encodes the polysaccharide β-1,6-GlcNAc; involved in cell-to-cell adherence and cell surface adherence; contributing to biofilm formation [16] within the cell-sputum model. Testing within a cell/sputum model can help uncover up or downregulation of similar genes as the bacteria have a live cell model to respond to during infection

Line 593 - 605:

Future work could involve assess continuation of testing in different other emergent bacterial species, like Stenotrophomonas maltophilia [17], and improving the model in which testing is performed. A. xylosoxidans mucin adhesion suggests an investigation into relative bacterial colonisation in the ASMDM-1 layer versus the adherent cell layer is warranted. This could be accomplished by fluorescently tagging the bacteria and performing live cell confocal microscopy to monitor colonisation and eventual biofilm formation [18-20]. The use of BEAS-2B cells and THP-1 cells provides insights into how these cells may response to bacterial infection especially as both as actively invovled in the CF proinflammatory response [19]. Therefore, aAnother future direction could be co-culture of THP-1 and BEAS-2B cells to investigate the cell-to-cell signalling and proinflammatory [21] responses to bacterial infection thus shedding light on an investigate how CT either mitigates or abrogates cell processes. Studies by Li et al. (2012), assessed the effect of asbestos particles in transwells on the proinflammatory responses of THP-1 cells, in the apical chamber, and their effects on bronchial epithelial cells, in the basolateral chamber [21]. This model may be altered to suit specificASMDM-1 requirements and could highlight the impact of cytokine responses and oxidative stress an antioxidant effect on each cell.

  1. xylosoxidans is mostly written in italics, but not on page 3 (thrice) and p11. On p12, in the heading, the bacterium is underscored, perhaps to differentiate from the rest of the text in italics; usually bacterial names are written as standard text, to differentiate from letters in italics.

Response: Thank you for your comment. All instances of A. xylosoxidans in text have been checked and italicised with the following exceptions:

Line 131: A. xylosoxidans….(written as standard text in italicised title)

Line 376: A. xylosoxidans….(written as standard text in italicised title)

Line 405: A. xylosoxidans….(written as standard text in italicised title)

---------------------------------------------- 

1. Garrigos, T., et al., Distribution of Achromobacter Species in 12 French Cystic Fibrosis Centers in 2020 by a Retrospective MALDI-TOF MS Spectrum Analysis. J Clin Microbiol, 2022. 60(6): p. e0242221.

2. Garrigos, T., et al., Development of a database for the rapid and accurate routine identification of Achromobacter species by matrix-assisted laser desorption/ionization-time-of-flight mass spectrometry (MALDI-TOF MS). Clin Microbiol Infect, 2021. 27(1): p. 126.e1-126.e5.

Reviewer 2 Report

This paper describes the creation of a model for lung infection in cystic fibrosis patients. There were three stages to the testing. 1), The tolerance of BEAS-2B cell lines and two A. xylosoxidans strains to the ASMDM-1 medium were optimized. 2), the cytotoxicity of the combined therapy that combined N-acetylcysteine with an antibiotic was tested, on cells alone in both BEAS-2B cells (human bronchial epithelium) and THP-1 cells (representing lung macrophages). 3), the efficacy of the combined therapy was tested on a bacterial infection using the live cells in the sputum model. They found that a combined therapy using colistin was tolerated well by the cells and significantly reduced the bacterial count. The development of the model is an important advance and a valuable contribution. The paper is well-written and organized. I recommend publication.

Author Response

  1. The paper describes the creation of a model for lung infection in cystic fibrosis patients. There were three stages to the testing. 1), The tolerance of BEAS-2B cell lines and two A. xylosoxidans strains to the ASMDM-1 medium were optimized. 2), the cytotoxicity of the combined therapy that combined N-acetylcysteine with an antibiotic was tested, on cells alone in both BEAS-2B cells (human bronchial epithelium) and THP-1 cells (representing lung macrophages). 3), the efficacy of the combined therapy was tested on a bacterial infection using the live cells in the sputum model. They found that a combined therapy using colistin was tolerated well by the cells and significantly reduced the bacterial count. a

Response: We thank the reviewer for their kind comments and for their recommendation to publish.